# Digital transformation, dynamic capabilities and new quality productive forces: Empirical data from listed Chinese manufacturing companies

**Junling Wang[1], Yongjun Zhang[1], Ruifen Zhao[2]***

**1** Management School, Hebei GEO University, Shijiazhuang, China, **2** Economics School, Hebei GEO University, Shijiazhuang, China

* zhaoruifen_199@163.com

## Abstract

In the context of the rapid growth of the digital economy, it is of great significance to explore how digital transformation can promote the development of new quality productive forces (Nqpf). Drawing on data from A share manufacturing listed companies in Shanghai and Shenzhen between 2011 and 2022, this study empirically analyzes the effect of digital transformation in the manufacturing industry on emerging Nqpf. The results reveal that digital transformation in the manufacturing industry has a significant positive impact on enhancing the level of Nqpf. Mechanism tests show that digital transformation in the manufacturing industry can promote the growth of Nqpf by strengthening companies' innovation, absorptive, and adaptive capabilities. Heterogeneity tests further indicate that the influence of digital transformation on advancing Nqpf is particularly pronounced in Non-state-owned, High-tech, and Growth stage manufacturing companies. The research results uncover the underlying mechanisms through which digital transformation in the manufacturing industry influences Nqpf, and offering valuable theoretical and practical insights for manufacturing companies on how to leverage digital transformation to promote the development of Nqpf.

## 1. Introduction

Since July 2023, General Secretary Xi Jinping has introduced the concept of "New Quality Productive Forces (Nqpf)" during his inspections across multiple regions. Unlike traditional labour-intensive and resource-dependent productivity paradigms, this new form relies more heavily on factors that advance innovation, knowledge, technology, and data, emphasizing efficient and green development patterns. Nqpf fundamentally represent an innovative form of productivity. It drives the creative reallocation of production factors and facilitates the profound transformation and upgrading of industries by leveraging scientific and technological advancements. It is characterized by new quality labourers with strong innovation awareness and high professional qualities, data driven new quality means of labour, new quality

**Data availability statement:** All relevant data are within the manuscript and its Supporting Information files.

**Funding:** This work was supported by the Hebei Federation of Social Sciences [grant 202307013 to JW]; Hebei Provincial Department of Education [grant 2402013 to RZ]; And Shijiazhuang Science and Technology Bureau [grant 245790125A to JW].

**Competing interests:** we have no known competing financial interests or personal relationships that could have appeared to influence the work reported in this paper.

objects of labour featuring industrial innovation and environmental friendliness, and the qualitative change resulting from the optimized combination of the three [1]. Nqpf not only provide a new driving force for economic development, but also are the key force for realizing national modernization, enhancing international competitiveness, and promoting comprehensive social progress. In recent years, in order to achieve economic recovery, countries around the world have begun to vigorously develop advanced manufacturing. However, during the process of transitioning from scale expansion to quality and efficiency improvement, manufacturing enterprises still face prominent issues such as "low efficiency, high consumption, and high emissions". Against this backdrop, cultivating Nqpf has emerged as a crucial path for the manufacturing industry to break through bottlenecks and achieve high quality development. However, the overall level of Nqpf in the manufacturing sector remains relatively low at present, leaving considerable room for growth. Meanwhile, the transformation centered on digital transformation has provided new opportunities for cultivating Nqpf. On the one hand, the profound integration and extensive application of cutting-edge technologies like artificial intelligence, big data, and cloud computing promote the restructuring and optimization of industrial frameworks, opening up new business models and service methods. For example, in the field of automobile manufacturing, Xiaomi Auto has achieved highly efficient automated production through its smart factory, with one SU7 rolling off the production line every 76 seconds. In the field of electronics manufacturing, the Electronic Works Amberg (EWA) has realized full process data collection and intelligent management and control with the help of the MindSphere industrial Internet of Things operating system, achieving a product qualification rate of 99.9988%. On the other hand, the strategic and future industries spawned by digital technologies provide a strong foundation for the in depth adjustment of the economic structure [2]. For instance, embodied intelligent robots and the quantum information industry are emerging as crucial sectors that will lead economic development and enhance a nation's core competitiveness in the future. Therefore, researching the impact of digital transformation on Nqpf holds substantial importance not only for the high-quality growth of manufacturing enterprises but also for nations' advancement in developing sophisticated manufacturing industry.

Currently, domestic scholars have conducted multi dimensional explorations on the correlation between "digital transformation and Nqpf". In terms of qualitative research, Zhai and Pan (2024) [3] further analyzed the internal mechanism of interaction between digital transformation and Nqpf from the framework of "dynamics - elements - structure", revealing the profound impact of digital transformation. Meanwhile, some studies have also pointed out the positive role of digital transformation in promoting the in depth integration of the digital and real economies, the transformation of production modes, and the optimal allocation of resources [4]. Nevertheless, it also encounters various challenges, including delays in the innovation of the digital economy system, the imperfect science and technology management system, the shortage of talents, and data security and privacy protection [5,6]. In terms of quantitative research, scholars have verified the positive influence of digital transformation on Nqpf through empirical analysis. Specifically, digital transformation enhances the

growth of Nqpf through various channels, including improving the quality of enterprise internal control [7], promoting technological and management innovation [8], alleviating financing constraints [9,10], and enhancing supply chain resilience [11]. In addition, some studies have found the "U" characteristic in how digital transformation influences Nqpf [12], enriching the dynamic effect in how digital transformation influences Nqpf. The aforementioned studies have examined the connection between digital transformation and Nqpf from different perspectives, providing theoretical insights and empirical experiences for understanding how digital transformation empowers the Nqpf. However, there are still some deficiencies. Firstly, at the theoretical level, there is a need for more profound analysis of the relationship in how digital transformation influences Nqpf, in order to better understand the underlying interaction mechanisms between them. Secondly, current research lacks sufficient empirical validation of the underlying mechanisms, with most studies concentrating on listed companies as the primary subject, thereby overlooking unique attributes and characteristics of different sectors.

The incorporation of digital technologies and the resulting corporate innovation activities essentially rely on certain resources and capabilities for support [13]. Therefore, dynamic capabilities are expected to serve as an essential intermediary mechanism for analysing how digital transformation empowers the advancement of Nqpf. Firstly, digital transformation reconstructs the enterprise's resource base through digital technologies, providing technical support for the development of dynamic capabilities. Specifically, enterprises can cultivate innovation capability by accelerating technological breakthroughs and model innovation driven by data, enhance absorptive capability by efficiently integrating external knowledge through platforms such as the industrial Internet, and improve adaptive capability by quickly responding to environmental changes through intelligent systems. In addition, enterprises, relying on dynamic capabilities, promote the in depth integration of new elements such as data and computing power with traditional resources, achieve an intelligent and green production state, and thus drive the advancement of Nqpf. Therefore, drawing on the Resource-Based View and the dynamic capabilities theory, this study selects manufacturing A share listed companies in Shanghai and Shenzhen from 2011 to 2022 as the research sample. It aims to investigate the influence of digital transformation on the advancement of Nqpf and the mediating role of dynamic capabilities, with the object of offering theoretical and practical references for the decision making of manufacturing transformation, upgrading, and high quality development.

The possible contributions of this research can be summarized as follows: Firstly, it integrates digital transformation, dynamic capabilities, and Nqpf into a theoretical framework, clarifying the mechanism through which digital transformation impacts Nqpf. Secondly, it expands the application scenarios of the dynamic capabilities theory, extending from traditional environmental adaptability research to the field of technology driven productivity transformation. Thirdly, it constructs a three dimensional indicator system encompassing new quality labourers, new quality means of labour, and new quality objects of labour, providing a reference for the operational measurement of the Nqpf level in the manufacturing industry.

## 2. Theoretical analysis and research hypotheses

### 2.1. Digital transformation of manufacturing and developm- ent of Nqpf

The digital transformation of the manufacturing industry can promote the development of Nqpf, which is mainly shown in the role of digital technologies in assisting enterprises to acquire heterogeneous resources. The Resource-Based View emphasizes that an enterprise is a aggregation of resources, and the heterogeneity of resources is the key factor explaining the differences in enterprise performance [14]. Digital transformation is defined as a change process initiated by the infrastructure of digital technologies, digital products and digital platforms, which ultimately leads to changes at multiple levels, including individuals [15], organisations [16] and industries [17]. In essence, it reconstructs the allocation mode of production factors through digital technologies and transforms traditional resources into a new type resource system characterized by advanced technology, efficiency, and quality. Digital transformation brings valuable, scarce, and non-substitutability digital resources to enterprises [18], including digital technologies and data information. These resources serve as an important foundation for businesses to develop Nqpf. Specifically, digital transformation empowers the three elements of Nqpf (new quality labourers; new quality means of labour; new quality objects of labour), triggers qualitative

changes in production factors, optimizes and upgrades their combination, and thus promotes the transformation and advancement of the manufacturing sector towards intelligence, green development, and high efficiency.

In terms of the new quality labourers, the digital transformation of the manufacturing sector can reshape workers' roles, knowledge, and skill sets, thereby facilitating the development of Nqpf. On the one hand, a series of emerging occupational roles (such as cloud-network intelligent operation and maintenance technicians and generative artificial intelligence system application technicians) have emerged during the digital transformation, becoming a key driving force for industrial innovation and upgrading. On the other hand, the digital transformation places higher requirements on the quality of workers. Workers are not only required to be proficient in traditional manufacturing processes and skills but also need to deeply integrate cutting edge technological knowledge in information technology, data analysis, artificial intelligence [19], to calmly handle and manage the intelligent and automated modern production environment. The evolving trend of skill compounding encourages workers to engage in continuous learning and professional training, continuously enhancing their comprehensive qualities and innovation capability, thus laying a solid foundation of high skilled talents for the development of Nqpf.

In terms of new quality means of labour, the digital transformation of the manufacturing sector can intelligently upgrade the means of labour and promote the development of Nqpf. As the material foundation in the manufacturing production process, the digital transformation of the means of labour is primarily showed in the intelligent upgrading of production equipment and the optimization and reorganization of production processes. By introducing advanced technologies that include intelligent robots, automated processing lines, and the Internet of Things, it is possible to achieve precise control and efficient management of the production process, thereby enhancing the efficiency of production and the quality of the product [20]. Meanwhile, digital transformation facilitates the digital management of means of production. For example, technologies like cloud computing and big data are used to collect and analyze production data in real time, providing a scientific basis for production decision – making and further optimizing resource allocation and production planning [21]. The intelligent upgrading of the means of labour not only reduces production costs but also enhances the flexibility and response speed of the manufacturing industry, providing a robust material fundament for the development of Nqpf.

In terms of new quality objects of labour, the digital transformation of the manufacturing sector can innovate and upgrade labour objects, thereby promoting the development of Nqpf. As the direct target of manufacturing production activities, the digital transformation of labour objects is reflected in the innovation and upgrade of products and the reconstruction of their value. To be specific, digital transformation allows the manufacturing industry to respond more rapidly to market demands. Through methods such as personalized customization and flexible production, it can offer consumers a wider variety of product choices, meeting consumers' demands for high quality and personalized products [11]. Meanwhile, digital transformation drives the manufacturing industry to transform into service – oriented manufacturing. By providing value added services such as remote operation and maintenance and intelligent diagnosis, it extends the product value chain and enhances the addition value of products. The innovation and upgrade of labour objects and the reconstruction of their value can enhance the market competitiveness of the manufacturing sector and create new growth points for the development of Nqpf.

In terms of the ecological environment, digital transformation promotes the greening of the industry, contributing to the development of Nqpf. Digital transformation enhances resource utilization efficiency and reduces energy waste and carbon emissions through precise monitoring and intelligent scheduling of the production process. Meanwhile, it promotes transparency and collaboration in the supply chain, constructs a green supply chain, ensures the green procurement of raw materials, and drives the green transformation of the entire supply chain. Additionally, the establishment of digital regulatory platforms and environmental information disclosure mechanisms effectively strengthens environmental supervision, enhances enterprises' awareness of environmental protection and sense of responsibility, and contributes to a mutually beneficial outcome in terms of economic and environmental performance.

According to the above theoretical analysis, this study proposes the following hypotheses:

H$_1$: The digital transformation of the manufacturing industry can promote the development of Nqpf.

## 2.2. The mediating role of dynamic capabilities

Dynamic capabilities denote an enterprise's capability to integrate, establish, and reconfigure both internal and external resources in response to external environmental changes, enabling the firm to sustain or gain competitive advantages in a constantly evolving business landscape [22]. They play a significant role in improving an enterprise's performance [23,24]. In the digital age, artificial intelligence is also regarded as a high order dynamic capability [25], enabling enterprises to effectively respond to the continuously evolving external environment, better allocating resources, and promoting the development of Nqpf [26]. Considering that digital transformation is a process of adapting to drastic changes in the external environment such as the influence of digital technologies and the upgrading of market demands, and that Nqpf is centered around innovation driven development. this study adopts the dynamic capabilities theoretical framework proposed by Wang and Ahmed (2007) [27], which is more relevant to the research content, to analyze the influence mechanism of digital transformation in the manufacturing sector on Nqpf. This framework divides dynamic capabilities into three core dimensions: innovation capability, absorptive capability, and adaptive capability, and this division has been widely recognized and verified in the academic community. Specifically, innovation capability is defined as an enterprise's capability to create novel value and drive strategic transformation by developing new products, services, or business models; absorptive capability emphasizes an enterprise's capability to identify and obtain external knowledge and to digest and transform it into resources that are then available to the enterprise.; adaptive capability refers to an enterprise's capability to rapidly modify its strategic direction, resource distribution, and organisational processes in reaction to the external environment. Digital transformation is viewed as a significant driver that accelerates the development and enhancement of a company's dynamic capabilities [28]. Meanwhile, dynamic capabilities are the cornerstone for an enterprise to seize competitive advantages [29], and Nqpf is the most core and direct manifestation of this competitive advantage.

Firstly, the digital transformation of the manufacturing sector promotes the development of Nqpf by enhancing innovation capability. The application of digital technologies enables firms to build powerful data processing and analysis capabilities, allowing them to more accurately capture changes in market demand and predict industry trends, thereby inspiring innovation and effectively avoiding the "innovation trap" caused by conservatism [30]. Meanwhile, digital transformation promotes cross domain and cross industry cooperation and integration, bringing broader innovation perspectives and more innovation resources to enterprises [31]. By constructing an open innovation ecosystem, enterprises can more effectively integrate internal and external innovation elements, accelerate the R&D and commercialization processes of new technologies and new products, and comprehensively enhance Nqpf.

Secondly, the digital transformation of the manufacturing sector promotes the development of Nqpf by enhancing absorptive capabilities. Digital transformation breaks the boundaries between firms and between firms and the external environment, facilitating information sharing and collaborative cooperation among enterprises and reducing the barriers and costs of knowledge transfer [32]. Relying on digital platforms, enterprises can instantly absorb knowledge sharing from stakeholders such as collaborators, consumers, and competitors, achieve in depth interaction and learning of internal and external information, and jointly develop new technologies and new products, thereby accelerating the process of knowledge absorptive and transformation. In addition, digital transformation enhances employees' acceptance and application capabilities of new knowledge and new technologies by improving their digital skills and literacy.

Finally, the digital transformation of the manufacturing sector promotes the development of Nqpf by enhancing adaptive capability Digital transformation involves multi dimensional upgrades and innovations in enterprise production processes, organizational structures and risk management [33]. First of all, the implementation of digital technologies facilitates firms to respond more rapidly to market variations, adjust production plans and product strategies. By constructing flexible

production systems and supply chain networks, enterprises can achieve instant response and precise satisfaction of market demand. In addition, digital transformation promotes the flattening and flexibility of enterprise organizational structures, improving the decision – making efficiency and execution ability of enterprises. Finally, through real time monitoring and analysis of market data and operational data, enterprises can promptly identify potential risks and take effective measures for prevention and response, enhancing their risk early warning and response capabilities. The comprehensive innovation brought about by digital transformation gradually strengthens enterprises' adaptive capability to the environment, which is a solid guarantee for Nqpf.

According to the above theoretical analysis, this study proposes the following hypotheses:

$H_{2a}$: Innovation capability play an intermediary role in the process where the digital transformation of the manufacturing sector promotes the development of Nqpf.

$H_{2b}$: Absorptive capability play an intermediary role in the process where the digital transformation of the manufacturing sector promotes the development of Nqpf.

$H_{2c}$: Adaptive capability play an intermediary role in the process where the digital transformation of the manufacturing sector promotes the development of Nqpf.

In summary, this paper constructs the following research model, as shown in Fig 1.

## 3. Research design

### 3.1. Sample selection and data sources

Given that the rapid development of China's mobile Internet commenced after 2011, this study selects manufacturing A share listed companies in Shanghai and Shenzhen from 2011 to 2022 as the research sample. To ensure the research quality, we excluded samples in abnormal states including ST, *ST, and PT, also excluded abnormal or severely missing key variable data. Additionally, a 1% winsorisation is employed for all continuous variables, reducing the impact of possible outliers on the research outcomes. After the above mentioned processes, this study achieved 20,986 valid observations.

The data sources for the main variables are as follows: The characteristic words for measuring digital transformation, along with the data for measuring dynamic capabilities such as R&D investment, the number of technical personnel, total operating revenue, and enterprise R&D, capital, and advertising expenditures, are all sourced from the annual reports of the sample firms. The data for the environmental governance level indicator, which measures Nqpf, is obtained from the Huazheng ESG Rating System, and the data for other indicators also come from the enterprises' annual reports. The data for control variables such as enterprise age, enterprise size, and board size are derived from the CSMAR and WIND databases.

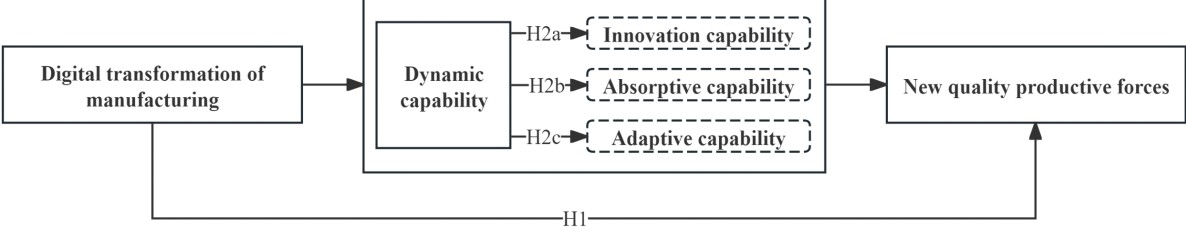

**Fig 1. Research Model.**

### 3.2. Variable definition and measurement

**3.2.1. Dependent variable: New quality productive forces (Nqpf).** Currently, labourers in China's manufacturing sector are undergoing a profound transformation from traditional skilled labourers to those with high quality and high skills. They not only need to be proficient in professional skills but also possess innovative thinking and the ability for continuous learning. Secondly, as the material foundation of production activities, labour materials, empowered by digital technology, exhibit characteristics of high tech, intelligence, and automation. They are gradually replacing the traditional and backward production equipment in the manufacturing industry, enabling efficient operation of the production process and intelligent management. Finally, with the constantly changing market needs and continuous technological progress, the manufacturing industry is actively expanding diversified labour objects such as new materials and intelligent products to meet new market demands. Meanwhile, the selection of green and sustainable labour objects has also become an influential trend in the advancement of Nqpf in the manufacturing industry. Therefore, referring to existing research [34,35], this paper constructs constructsa a system of evaluation indices for the level of Nqpf in the manufacturing industry from three dimensions: new quality labourers, new quality means of labour and new quality objects of labour, and calculates the total score using the entropy weight method, as shown in Table 1.

First, this paper measures new quality labourers in two dimensions: cultural quality and professional quality. Among these, the proportion of highly educated personnel reflects the cultural quality of new quality labourers. The digital background of executives, the proportion of R&D personnel, and the salary proportion of R&D personnel reflect the professional quality of new quality labourers. Second, new quality means of labour are measured from material and nonmaterial labour materials. Among these, the proportion of fixed assets is used to measure material labour materials, reflecting the reduced dependence of the manufacturing industry on traditional assets under digital transformation. To measure the risk resilience, innovation ability, and operational ability of the manufacturing industry, seven indicators are selected: the proportion of R&D depreciation and amortization, the proportion of R&D leasing expenses, the proportion of direct R&D investment, the proportion of intangible assets, enterprise innovation output, total asset turnover, and the reciprocal of the equity multiplier. Third, new quality objects of labour are measured from digital business and the ecological environment. Among these, the proportion of digital assets and the proportion of manufacturing expenses reflect the growth of digital business in the manufacturing sector under digital empowerment. Environmental governance levels and executives' green awareness reflect the manufacturing industry's positive contributions to the ecological environment.

**3.2.2. Independent variable: Digital transformation (Dit).** This study adopts the feature-word map of firm digital transformation constructed by Wu et al. (2021) [36], which encompasses two aspects: the application of four types of underlying technologies, namely artificial intelligence technology, big data technology, cloud computing technology, and blockchain technology, and the practical application of these technologies. Meanwhile, based on the Python software, a search is conducted for digital feature words in the annual report texts of manufacturing listed companies on the Shanghai and Shenzhen A share markets from 2011 to 2022. Subsequently, the word frequencies are aggregated and logarithmically processed, ultimately yielding the overall indicator of digital transformation for manufacturing listed companies. The structured feature-word map of firm digital transformation adopted in this research is shown in Fig 2.

**3.2.3. Mediating variables: Innovation capability (IA), absorptive capability (AP), and adaptive capability (AC).** According to the research of Zhao et al. (2016) [37] and Yang et al. (2020) [38], dynamic capabilities are classified into three dimensions: innovation capability (IA), absorptive capability (AP), and adaptive capability (AC). Innovation capability is assessed using the sum of a comprehensive evaluation using two metrics: R&D investment and the percentage of technical staff. Absorptive capability assessed using by the ratio of R&D investment to total operating revenue. Adaptive capability is assessed using the negative of the coefficient of variation calculated from the intensity of expenditures on enterprise R&D, capital, and advertising. Among them, R&D expenditure intensity is measured by the ratio of R&D expenditure to total operating income; capital expenditure intensity is measured by the ratio of cash paid for the purchase and construction of fixed assets, intangible assets, and other long-term assets to total operating income;

**Table 1. Evaluation index system of Nqpf in the manufacturing industry.**

| Target level | Criterion level | First-level indicator | Second-level indicator | Indicator description |
|---|---|---|---|---|
| Nqpf level of manufacturing industry | New quality labourers | Cultural quality | Proportion of highly educated personnel | Number of personnel with bachelor's degree or above/ Total number of employees |
| | | Professional quality | Digital background of executives | Executives with digital background are assigned a value of 1, otherwise 0 |
| | | | Proportion of R&D personnel | Number of R&D personnel/ Total number of employees |
| | | | Salary proportion of R&D personnel | R&D expenses – salaries and wages/ Total operating income |
| | New quality means of labour | Physical means of labour | Proportion of fixed assets | Proportion of fixed assets |
| | | Nonphysical means of labour | Proportion of R&D depreciation and amortization | R&D expenses – depreciation and amortization/ Total operating income |
| | | | Proportion of R&D leasing expenses | R&D expenses – leasing expenses/ Total operating income |
| | | | Proportion of direct R&D investment | R&D expenses – direct investment/ Total operating income |
| | | | Proportion of intangible assets | Intangible assets/ Total assets |
| | | | Enterprise innovation output | ln(Total number of patent applications in the current year +1) |
| | | | Total asset turnover ratio | Total operating income/ Average total assets |
| | | | Reciprocal of equity multiplier | Owner's equity/ Total assets |
| | New quality objects of labour | Digital business | Proportion of digital assets | Digital assets/ Intangible assets |
| | | | Proportion of manufacturing expenses | (Operating cash outflow + depreciation of fixed assets + amortization of intangible assets + impairment provision-purchasing goods to accept labour worth cash-paying wages to employees)/ (Operating cash outflow + depreciation of fixed assets + amortization of intangible assets + impairment provision) |
| | | Ecological environment | Environmental governance level | E Indicator Score in Huazheng ESG Rating System |
| | | | Green awareness of executives | ln(Frequency of green development keywords in annual report +1) |

**Note:** ①Digital background of executives: A company's directors, supervisors, or senior managers are considered to have a digital background if their specialized majors involve fields such as "Information, Intelligence, Software, Electronics, Communication, Systems, Networks, Automation, Wireless, and Computer." ②Operating cash outflow: The total amount of cash actually paid for the enterprise's operating activities during the current period. ③Depreciation of fixed assets: Non-cash expenses accrued for the current consumption of fixed assets. ④Amortization of intangible assets: Non-cash expenses accrued for the current consumption of intangible assets. ⑤Impairment provision: Asset impairment losses recognized in the current period, such as inventory write-downs and impairment of fixed assets. ⑥Purchasing goods to accept labour worth cash: Cash paid by the enterprise for the purchase of raw materials and other items during the current period. ⑦Paying wages to employees: The total amount of cash paid to employees by the enterprise in the current period, including salaries, bonuses, and allowances. ⑧Green development keywords: Based on three dimensions—green competitive advantage awareness, corporate social responsibility awareness, and perception of external environmental pressure—the following keywords are selected: energy conservation and emission reduction, environmental protection strategy, environmental protection concept, environmental management organization, environmental protection education, environmental protection training, environmental technology development, environmental audit, energy saving and environmental protection, environmental protection policy, environmental protection department, environmental protection inspection, low carbon and environmental protection, environmental protection work, environmental protection governance, environmental protection and environmental governance, environmental protection facilities, environmental protection related laws and regulations, and environmental protection and pollution control.

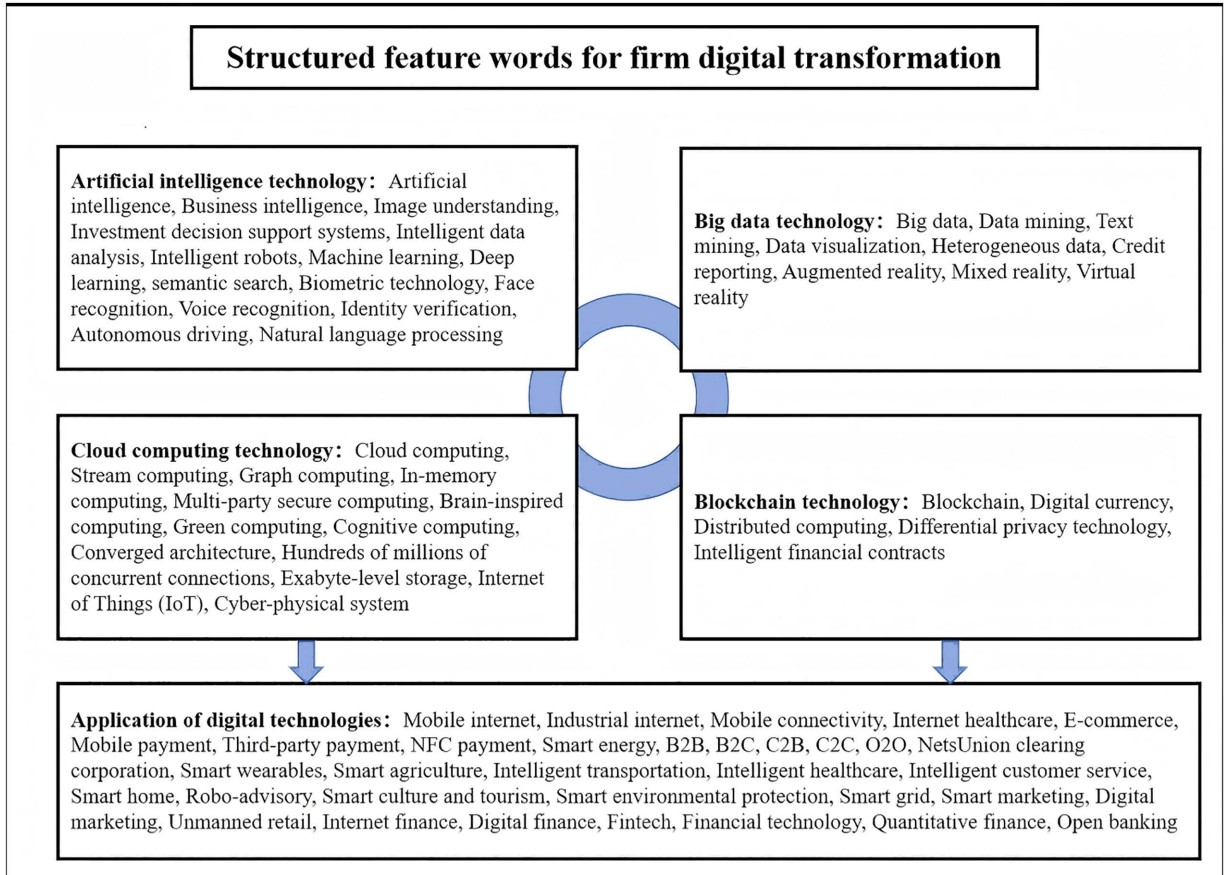

**Fig 2. The structured feature-word map of firm digital transformation.**

and advertising expenditure intensity is measured by the ratio of advertising and promotion expenses to total operating income.

   **3.2.4. Control variables.** To eliminate the influence of other factors on the research results, this research includes the following variables: firm age (Age), firm size (Size), board size (Board), total asset growth rate (Growth), current asset ratio (Liquid), asset-liability ratio (Leverage), executive team size (TMunm), independent director ratio (Indep), ownership concentration (Top1), industry dummy variables (Industry), and year dummy variables (Year).

   Definitions of the key variables are presented in Table 2.

## 3.3. Research model construction

To examine the influence of digital transformation in manufacturing, dynamic capabilities, and Nqpf, this research follows the relevant research practices [39] and first constructs model (1) to test the main effects.

$$Npro_{i,t} = \alpha_0 + a_1 Dit_{i,t} + \alpha_2 Controls + \sum Industry + \sum Year + \varepsilon_{i,t} \tag{1}$$

In formula (1): $Dit_{i,t}$ denotes the degree of digital transformation of enterprise i in year t; $Nqpf_{i,t}$ denotes the level of new quality productive forces of enterprise i in year t; Controls denotes the all control variables; $\sum Industry$ and $\sum Year$ denotes the industry and time fixed effects respectively; and $\varepsilon_{i,t}$ denotes the error term.

**Table 2. Definition of the main variables.**

| Variable type | Variable name | Variable symbol | Variable measurement |
|---|---|---|---|
| Dependent variable | New quality productive forces | Nqpf | According to the three components of productivity, the evaluation index system is constructed and calculated by entropy weight method. |
| Independent variable | Digital transformation | Dit | The number of keywords involved in digital transformation in the annual report of the enterprise plus 1 natural logarithm |
| Mediating variables | Innovation capability | IA | The R&D investment and the percentage of technical staff are added after standardized treatment. |
| | Absorptive capability | AP | R&D investment/ Total operating income |
| | Adaptive capability | AC | - (standard deviation of R&D investment intensity, capital expenditure intensity and advertising expenditure intensity)/ (Mean value of R&D investment intensity, capital expenditure intensity and advertising expenditure intensity) |
| Control variables | Firm age | Age | ln (current year-year of establishment + 1) |
| | Firm size | Size | The total assets of the company take the natural logarithm. |
| | Board size | Board | Natural logarithm of the number of board members |
| | Total asset growth rate | Growth | (Total assets at the end of the period-total assets at the beginning of the period)/ Total assets at the beginning of the period |
| | Current asset ratio | Liquid | Current assets/ Current liabilities |
| | Assets-liability ratio | Leverage | Total liabilities/ Total assets |
| | Executive team size | TMnum | The number of executives takes the natural logarithm |
| | Independent director ratio | Indep | Number of independent directors/ Number of board of directors |
| | Ownership concentration | Top1 | Measurement of the shareholding ratio of the largest shareholder |
| | Industry | Industry | Industry fixed effect |
| | Year | Year | Year fixed effect |

Secondly, based on Model (1), this paper constructs Model (2) and Model (3) to examine the mediating effect of dynamic capabilities.

$$Mediator_{i,t} = \beta_0 + \beta_1 Dit_{i,t} + \beta_2 Controls + \sum Industry + \sum Year + \varepsilon_{i,t} \tag{2}$$

$$Npro_{i,t} = \gamma_0 + \gamma_1 Dit_{i,t} + \gamma_2 Mediator_{i,t} + \gamma_2 Controls + \sum Industry + \sum Year + \varepsilon_{i,t} \tag{3}$$

In formulas (2) and (3): $Mediator_{i,t}$ represents the dynamic capabilities of enterprise i in year t, which are respectively expressed by innovation capability (IA), absorption capability (AP) and adaptive capability (AC), and other variables are the same as above.

## 4. Empirical analysis

### 4.1. Descriptive statistical analysis

Table 3 presents the descriptive statistical results of the main variables. Among them, for the explained variable Nqpf, the mean value is 1.527, the median is 0.986, and the standard deviation is 1.855. In addition, there is a significant difference between the highest and lowest values, showing that the development of Nqpf in the manufacturing industry is unbalanced, and most enterprises are at a low level of Nqpf. Therefore, the Nqpf of China's manufacturing sector has promising growth prospects. For the core explanatory variable Dit, the mean value is 1.300, the median is 1.099, and the standard

**Table 3. Descriptive statistical results of the main variables.**

| Variable name | Sample size | Mean value | Minimum value | Maximum value | Standard deviation | Median |
|---|---|---|---|---|---|---|
| Nqpf | 20986 | 1.527 | 0.276 | 8.602 | 1.855 | 0.986 |
| Dit | 20986 | 1.300 | 0.000 | 5.011 | 1.279 | 1.099 |
| IA | 20986 | 0.371 | 0.000 | 1.630 | 0.252 | 0.308 |
| AP | 20986 | 0.047 | 0.000 | 0.310 | 0.044 | 0.038 |
| AC | 20986 | −1.057 | −1.732 | −0.264 | 0.314 | −1.015 |
| Age | 20986 | 2.912 | 1.386 | 3.526 | 0.326 | 2.944 |
| Size | 20986 | 22.097 | 19.676 | 26.086 | 1.172 | 21.928 |
| Board | 20986 | 2.110 | 1.609 | 2.708 | 0.190 | 2.197 |
| Growth | 20986 | 0.168 | −0.294 | 2.589 | 0.326 | 0.093 |
| Liquid | 20986 | 2.657 | 0.316 | 17.317 | 2.591 | 1.798 |
| Leverage | 20986 | 0.393 | 0.052 | 0.894 | 0.191 | 0.384 |
| TMnum | 20986 | 1.781 | 0.693 | 2.639 | 0.363 | 1.792 |
| Indep | 20986 | 0.376 | 0.143 | 0.571 | 0.054 | 0.333 |
| Top1 | 20986 | 0.333 | 0.086 | 0.743 | 0.141 | 0.311 |

deviation is 1.279, showing a normal distribution, which meets the research conditions. Meanwhile, there is a large difference between the highest and lowest values, and the mean value is much smaller than the highest value, showing that the digital transformation speed of the manufacturing industry varies, and there are also clear distinctions in the degree of transformation. The descriptive statistical results of the remaining variables are similar to those of relevant studies. At the same time, the VIF among the main variables are all between 1.030 and 2.140, suggesting that there is no major issue of multicollinearity in the research.

## 4.2. Benchmark regression analysis

The results of the benchmark regression of the digital transformation on the level of Nqpf are presented in Table 4. Column (1) shows the direct regression results of digital transformation on Nqpf; Columns (2) and (3) display the regression findings after additionally accounting for year and industry fixed effects and incorporating control variables; Column (4) presents the regression outcomes when both year and industry fixed effects are considered, along with the inclusion of control variables. According to the above findings, it can be observed that the Dit coefficients of the four regression methods are all significantly positive at the 1% level, showing that the digital transformation of the manufacturing industry has a significantly positive impact on the development of Nqpf, thus validating Hypothesis H1 of this research.

## 4.3. Robustness test

To strengthen the reliability of the research findings, this research employs multiple approaches for verification.

 **4.3.1. Replacement of the dependent variable.** During the 11th group study session of the Political Bureau of the CPC Central Committee, General Secretary Xi Jinping emphasized that Nqpf are characterized primarily by substantial improvements in total factor productivity (TFP). Therefore, this research utilizes TFP as an alternative measure of Nqpf, which are calculated using three distinct methods: fixed effects (FE), the generalized method of moments (GMM), and the Levinsohn Petrin (LP) method. Columns (1), (2), and (3) in Table 5 present respectively the regression findings between digital transformation in manufacturing and TFP measured by the FE, GMM, and LP methods. Among them, the Dit coefficient of the FE method is significantly positive at the 5% level, while the Dit coefficients of the other two methods are significantly positive at the 1% level. These results suggest that, even when the dependent variable is substituted, the

**Table 4. Benchmark regression results.**

| Variable Name | (1) Nqpf | (2) Nqpf | (3) Nqpf | (4) Nqpf |
|---|---|---|---|---|
| Dit | 0.353*** (14.46) | 0.294*** (11.02) | 0.347*** (14.12) | 0.272*** (10.42) |
| IA | | | −0.277*** (−3.22) | −0.368*** (−3.87) |
| AP | | | 0.051* (1.67) | 0.044 (1.42) |
| AC | | | −0.156 (−0.86) | 0.040 (0.22) |
| Age | | | 0.160*** (3.27) | 0.154*** (3.17) |
| Size | | | 0.052*** (3.68) | 0.044*** (3.23) |
| Board | | | 0.132 (0.79) | −0.079 (−0.46) |
| Growth | | | 0.611*** (8.24) | 0.640*** (8.75) |
| Liquid | | | 0.848 (1.35) | 1.106* (1.80) |
| Leverage | | | −0.141 (−0.63) | 0.008 (0.03) |
| Constant | 1.067*** (35.16) | 1.148* (1.81) | −0.501 (−0.65) | −0.644 (−0.65) |
| Sample size | 20986 | 20986 | 20986 | 20986 |
| Industry fixed effect | No | Yes | No | Yes |
| Year fixed effect | No | Yes | No | Yes |
| Adjusted $R^2$ | 0.059 | 0.087 | 0.083 | 0.112 |

**Note:** *, * *, * * * are significant at the levels of 10%, 5%, and 1%, respectively, and the values in the brackets are t values, the same as those in the table below.

variable Dit within the manufacturing sector still demonstrates a statistically significant positive impact on Nqpf, thereby reaffirming Hypothesis $H_1$.

**4.3.2. Replacement of the independent variable.** This researsh remeasures digital transformation in manufacturing using the Digital Transformation Index of Listed Companies from the CSMAR database, denoted $Dit_1$. The index is defined as the following: $0.3472 \times$ strategic guidance, $0.1620 \times$ technology-driven, $0.0969 \times$ organizational empowerment, $0.0342 \times$ environme -ntal support, $0.2713 \times$ digital outcomes, and $0.0884 \times$ digital applications. Column (4) in Table 5 suggest that the $Dit_1$ coefficient is 0.035 and strongly positive at the 1% level. This finding demonstrates that, even when the independent variable is substituted, digital transformation in manufacturing still significantly enhances Nqpf, providing additional support for Hypothesis $H_1$.

**4.3.3. Exclusion of outlier years.** This study excludes samples from 2015, 2016 (due to stock market crashes), and 2020 (due to the COVID-19 pandemic) to minimize the influence of external shocks on the findings. Column (5) in Table 5 reveals a Dit coefficient of 0.278, which remains strongly positive at the 1% level. These results confirm that, even after excluding outlier years, digital transformation in manufacturing has a significantly positive influence on Nqpf, further validating Hypothesis $H_1$.

**Table 5. Robustness test results.**

| Variable name | Replacement of dependent variable | | | Replacement of independent variable | Exclusion of outlier years | Lagged independent variable | Tobit model |
|---|---|---|---|---|---|---|---|
| | (1) TFP_FE | (2) TFP_GMM | (3) TFP_LP | (4) Nqpf | (5) Nqpf | (6) Nqpf | (7) Nqpf |
| Dit | 0.015** (2.14) | 0.069*** (7.87) | 0.050*** (6.67) | | 0.278*** (10.56) | | 0.090*** (7.88) |
| $Dit_1$ | | | | 0.035*** (9.09) | | | |
| L.Dit | | | | | | 0.269*** (9.64) | |
| Constant | −9.049*** (−32.58) | −0.750** (−2.40) | −5.960*** (−20.95) | −1.151 (−1.19) | −0.909 (−0.96) | −0.825 (−0.83) | 0.915 (1.59) |
| Control variables | Control | Control | Control | Control | Control | Control | Control |
| Sample size | 20303 | 20303 | 20303 | 20986 | 15909 | 17772 | 20986 |
| Industry fixed effect | Yes | Yes | Yes | Yes | Yes | Yes | Yes |
| Year fixed effect | Yes | Yes | Yes | Yes | Yes | Yes | Yes |
| Adjusted $R^2$ | 0.851 | 0.198 | 0.726 | 0.110 | 0.119 | 0.107 | – |
| LR chi2 | – | – | – | – | – | – | 902.931 |

**4.3.4. Lagged independent variable.** Considering potential lag effects in the influence of digital transformation on Nqpf, this study incorporates a one-period lagged digital transformation variable (denoted as L.Dit) into the regression model. Column (6) in Table 5 indicates that the L.Dit coefficient is 0.269 and strongly positive at the 1% level. The finding suggests that, even when lag effects are accounted for, digital transformation in manufacturing continues to significantly promote Nqpf, reinforcing the robustness of Hypothesis $H_1$.

**4.3.5. Change the measurement model.** To further examine the robustness of the findings, the research reexamines the influence of the digital transformation of the manufacturing sector on Nqpf using the Tobit model. According to the results in Column (7) of Table 5, the coefficient of Dit is 0.090, and it is strongly positive at the 1% level. The finding suggests that, even when the regression model is changed, the variable Dit within the manufacturing sector still demonstrates a statistically significant positive impact on Nqpf, and the conclusion of Hypothesis H1 in this research remains reliable.

## 4.4. Endogeneity test

To overcome potential issues of endogeneity, including sample self-selection, reverse causality and omitted variables, this paper employs the instrumental variable method and the propensity score matching (PSM) method for testing.

**4.4.1. Instrumental variable method.** Owing to the demonstration effect associated with enterprise digital transformation, the level of digital transformation among firms within the same province, industry, and time period is closely linked to the individual firm's digital transformation status [40], yet it has limited direct influence on the firm's Nqpf. Accordingly, this study follows the approach proposed by Xiao et al. (2021) [41], using the average level of digital transformation across the province, industry, and year (M_Dit) as the instrumental variable. The F test result is 2864.2, which is much greater than the critical value of 16.38 for weak instrumental variables, showing that there is no issue with weak instrumental variables. The two-stage least squares regression outcomes are presented in columns (1) and (2) of Table 6. In the first stage regression, the instrumental variable M_Dit shows a strongly positive association with digital transformation in the manufacturing sector, consistent with the expected research hypothesis. In the second stage

**Table 6. Endogeneity test results.**

| Variable name | Instrumental variable method | | PSM | |
|---|---|---|---|---|
| | (1)<br>The First Stage | (2)<br>The Second Stage | (3)<br>Calliper matching | (4)<br>Radius matching |
| | Dit | Nqpf | Nqpf | Nqpf |
| M_Dit | 0.996***<br>(53.52) | | | |
| Dit | | 0.474***<br>(13.09) | 0.282***<br>(9.61) | 0.272***<br>(10.41) |
| Constant | −2.226***<br>(−9.84) | −0.279<br>(−0.55) | −0.178<br>(−0.14) | −0.636<br>(−0.65) |
| Control variables | Control | Control | Control | Control |
| Sample size | 20986 | 20986 | 10029 | 20977 |
| Industry fixed effect | Yes | Yes | Yes | Yes |
| Year fixed effect | Yes | Yes | Yes | Yes |
| Adjusted $R^2$ | 0.277 | 0.097 | 0.108 | 0.112 |

estimation, the coefficient for Dit remains strongly positive at the 1% level. This indicates that, even when accounting for endogeneity, the positive impact of the digital transformation of the manufacturing industry on the level of Nqpf remains significant.

**4.4.2. Propensity score matching (PSM).** This research selects the PSM to avoid the sample self selection bias problem. In the model setting, referring to existing research [42], firms are categorized into a treatment group and a control group based on whether they have undergone digital transformation. Firms that have implemented digital transformation are classified into the treatment group and assigned a value of 1, while those that have not undergone such transformation are placed in the control group and coded as 0. The control variables in this paper are used as the covariates for matching variables, and the samples are matched respectively based on the caliper matching and radius matching principles. The results in Table 6 show that the coefficient of Dit is strongly positive at the 1% level under both caliper matching and radius matching, indicating that after eliminating the characteristic differences between digital transformation enterprises and non-digital transformation enterprises, the positive influence of digital transformation in manufacturing on the level of Nqpf remains significant.

## 5. Further analysis

### 5.1. Mediation effect test

Building upon the benchmark regression analysis conducted in the previous section, which revealed a positive relationship between digital transformation in the manufacturing sector and the development of Nqpf, we further investigate the mediating effect of dynamic capabilities using models (2) and (3). First, we performed a mediation analysis to assess the mediating role of innovation capability. The results are presented in columns (1) and (2) of Table 7. As shown, the regression coefficient of Dit on IA is 0.051, which is statistically significant at the 1% level. This suggests that digital transformation within the manufacturing sector contributes to an improvement in innovation capability. After adding Dit and IA to the regression model simultaneously, the coefficient of IA is 0.145, which is significantly positive at the 1% level. The coefficient of Dit is 0.205, which is slightly lower than the benchmark regression finding but still strongly positive at the 1% level. Second, we performed a mediation analysis to assess the mediating role of absorptive capability. The results are presented in columns (3) and (4) of Table 7. As shown, the regression coefficient of Dit on AP is 0.006, which is statistically significant at the 1% level. This suggests that digital transformation within the manufacturing sector contributes to

**Table 7. Test results of the mediating effect of dynamic capabilities.**

| Variable Name | (1) IA | (2) Npqf | (3) AP | (4) Npqf | (5) AC | (6) Npqf |
|---|---|---|---|---|---|---|
| Dit | 0.051*** (13.83) | 0.205*** (8.44) | 0.006*** (10.42) | 0.235*** (9.38) | 0.036*** (10.82) | 0.270*** (10.28) |
| IA | | 0.145*** (8.69) | | | | |
| AP | | | | 0.106** (2.07) | | |
| AC | | | | | | 0.128*** (7.32) |
| Constant | 0.459*** (4.99) | −1.255 (−1.28) | 0.092*** (6.45) | −1.203 (−1.21) | −1.561*** (−8.17) | −0.551 (−0.56) |
| Control variables | Control | Control | Control | Control | Control | Control |
| Sample size | 20986 | 20986 | 20986 | 20986 | 20986 | 20986 |
| Industry fixed effect | Yes | Yes | Yes | Yes | Yes | Yes |
| Year fixed effect | Yes | Yes | Yes | Yes | Yes | Yes |
| Adjusted $R^2$ | 0.274 | 0.136 | 0.254 | 0.127 | 0.107 | 0.112 |
| Bootstrap test | P=0.000 Confidence interval={0.226, 0.270} | | P=0.000 Confidence interval={0.258, 0.302} | | P=0.000 Confidence interval={0.320, 0.366} | |

an improvement in absorptive capability. After adding Dit and AP to the regression model simultaneously, the coefficient of AP is 0.106, which is strongly positive at the 5% level. The coefficient of Dit is 0.235, which is also slightly lower but still strongly positive at the 1% level. Finally, we performed a mediation analysis to assess the mediating role of adaptive capability. The results are presented in columns (5) and (6) of Table 7. As shown, the regression coefficient of Dit on AP is 0.036, which is statistically significant at the 1% level. This suggests that digital transformation within the manufacturing sector contributes to an improvement in adaptive capability. After adding Dit and AC to the regression model simultaneously, the coefficient of IA is 0.128, which is strongly positive at the 1% level. The coefficient of Dit is 0.270, which is also slightly lower but still strongly positive at the 1% level. The research finding suggest that innovation capacbility, absorptive capacbility, and adaptive capacbility all play a partial mediating role in the path by which the digital transformation of the manufacturing industry affects Nqpf. Hypotheses $H_{2a}$, $H_{2b}$, and $H_{2c}$ are verified, that is, the digital transformation of the manufacturing sector can foster the development of Nqpf by improving the levels of innovation capacbility, absorptive capacbility, and adaptive capacbility.

To ensure the robustness of the intermediate mechanism, this research employs the Bootstrap test approach to examine the above mentioned mediator paths. After 5,000 samplings, the mediating effect results of innovation capability, absorptive capability and adaptive capability are all P=0.000, and the confidence interval does not include 0, which once again proves the existence of the mediating influence of dynamic capabilities.

### 5.2. Heterogeneity test

To further confirm the promoting influence of digital transformation on Nqpf in manufacturing, this research examines the influence of digital transformation on Nqpf from three perspectives: the heterogeneity of manufacturing enterprises in terms of property rights, technological level, and life cycle. The findings are presented in Table 8.

Firstly, to explore whether the differences in the nature of enterprise ownership in the manufacturing industry would have different impacts on the development of Nqpf, the study the study categorizes the entire sample into state-owned enterprises and non-state-owned enterprises for heterogeneity analysis. The regression findings are shown in columns (1) and (2) of Table 8. The results show that the coefficients of Dit for both state-owned

**Table 8. Heterogeneity test results.**

| Variable name | Property rights | | Technological level | | Life cycle | | |
|---|---|---|---|---|---|---|---|
| | state-owned | non-state-owned | high-tech | non-high-tech | growth stage | mature stage | decline stage |
| | (1)<br>Nqpf | (2)<br>Nqpf | (3)<br>Nqpf | (4)<br>Nqpf | (5)<br>Nqpf | (6)<br>Nqpf | (7)<br>Nqpf |
| Dit | 0.237*** | 0.281*** | 0.231*** | 0.136*** | 0.283*** | 0.279*** | 0.260*** |
| | (13.40) | (21.72) | (13.12) | (10.40) | (12.65) | (18.76) | (13.26) |
| Constant | −0.971 | −1.189** | −0.932* | −1.765*** | −0.677 | −1.241* | 0.570 |
| | (−1.36) | (−2.00) | (−1.72) | (−4.05) | (−0.72) | (−1.84) | (0.70) |
| Control variables | Control | Control | Control | Control | Control | Control | Control |
| Sample size | 6083 | 14903 | 10493 | 10493 | 5241 | 10474 | 5271 |
| Industry fixed effect | Yes | Yes | Yes | Yes | Yes | Yes | Yes |
| Year fixed effect | Yes | Yes | Yes | Yes | Yes | Yes | Yes |
| Adjusted $R^2$ | 0.095 | 0.116 | 0.140 | 0.070 | 0.117 | 0.112 | 0.106 |

enterprises and non-state-owned firms are strongly positive at the 1% level, but the coefficient of Dit for non-state-owned firms is strongly greater than that for state-owned firms. This suggest that in the manufacturing industry, the digital transformation of non-state-owned firms has a stronger positive influence on the improvement of Nqpf than that of state-owned firms.

Secondly, as the core essence of Nqpf lies in technological innovation, to explore whether the level of technological advancement has a significant differential influence on the development of Nqpf during the digital transformation of manufacturing. This paper classifies all samples into high-tech and non-high-tech firms according to the 2012 industry classification of the China Securities Regulatory Commission and the "Key Areas of High and New Technology Supported by the State", and conducts group regression. From columns (3) and (4) of Table 8, it is evident that the coefficient of Dit for high-tech manufacturing enterprises is 0.231, significantly greater than 0.136 for non-high-tech manufacturing enterprises. This indicates that in manufacturing, the promotion v of digital transformation on Nqpf is more stronger positive in high-tech enterprises.

Thirdly, considering that enterprises have significant differences in investment decisions and strategic deployments at different life cycle stages [43]. Therefore, this paper further this paper further examines how the digital transformation of manufacturing firms affects the development of Nqpf at different life cycle stages. Following the approach of Li et al. (2011) [44], this study employs a composite score based on four indicators: sales revenue growth rate, retained earnings, capital expenditure, and enterprise age, to categorize the life cycle stages of firms. The firms with scores in the top 1/3 are classified as growth stage firms, those in the bottom 1/3 as decline stage firms, and the rest as mature stage firms. As presented in columns (5), (6), and (7) of Table 8, the Dit coefficients for manufacturing firms in the growth, mature, and decline stages are 0.283, 0.279, and 0.260 respectively, all of which are strongly positive at the 1% level. This suggests that digital transformation has the strongest positive impact on Nqpf development in firms during the growth stage, followed by those in the mature stage, and then those in the decline stage.

## 6. Discussion and conclusion

### 6.1. Research conclusions

Drawing on the Resource-Based View and dynamic capabilities theory, this research takes A share manufacturing listed companies in Shanghai and Shenzhen from 2011 to 2022 as the research objects, and examines the influence of digital transformation in manufacturing on the development of Nqpf and the mediating role of dynamic capabilities. The aim is

to offer theoretical and practical implications for the high quality development of manufacturing sector and the country's advancement of advanced manufacturing. The main research findings are as summarised below:

First, the empirical findings show that the digital transformation of the manufacturing industry has significantly enhanced the level of Nqpf. This implies that manufacturing enterprises can drive the intelligent transformation and upgrading of traditional industries, as well as foster the emergence of new industries through digital transformation, thereby comprehensively enhancing Nqpf. Although existing studies have researched the relationship between digital transformation and Nqpf, most of them rely on data from all listed companies [45,46], ignoring the unique industry attributes of the manufacturing sector such as labor intensity and environmental pollution [47]. Therefore, our research results provide new empirical evidence for the development of Nqpf in the manufacturing industry. Additionally, this research establishes a system of indices specifically designed to measure the level of Nqpf in manufacturing enterprises. Compared with the index systems constructed by Song et al. (2023) [48] and Li et al. (2024) [34], this research considers the "green" connotation of Nqpf and builds an ecological environment index, which can reflect the development of manufacturing enterprises in environmental protection. The research findings of this research not only expand the application context of digital transformation in promoting the development of Nqpf, but also contribute to the theoretical and empirical knowledge within this domain.

Second, the research findings reveal that the innovation capability, absorptive capability, and adaptive capability of enterprises play a partial mediating role in the process of promoting Nqpf through the digital transformation of the manufacturing industry. This implies that manufacturing enterprises can enhance their innovation capability, absorptive capability, and adaptive capability through digital transformation, thus facilitating the development of Nqpf. Previous studies have mostly focused on intermediary variables in corporate governance and finance, such as internal control [49], financing constraints [50,51], and corporate ESG performance [52,53]. However, this paper starts from the angle of dynamic capabilities, breaking through the limitations of the existing research framework and providing new pathways to advance Nqpf through the digital transformation of the manufacturing industry. In the meantime, the research results of this paper once again confirm that the digital transformation of firms is a dynamic process, and the capabilities of firms also change dynamically according to the internal and external environment [54]. The research incorporates digital transformation, dynamic capabilities, and Nqpf into a research framework, providing a solid theoretical basis for enterprises to cultivate dynamic capabilities and develop Nqpf in the process of digital transformation.

Third, the heterogeneity analysis reveals that digital transformation has a more pronounced influence on the Nqpf level in non-state-owned firm, high-tech firms. This finding aligns with the research results of Wang et al. [55] and Yuan et al. [56], who also emphasized how the property rights nature and technological level differences of manufacturing enterprises affect the development of Nqpf. Furthermore, this study builds on the work of Chin [26] and examines how digital transformation influences Nqpf by considering the heterogeneity across different stages of the firm life cycle. The study findings show that the more positive influence of digital transformation on Nqpf is more significant in manufacturing firm in the growth stage, followed by those in the mature stage, and finally those in the decline stage. The possible reasons are that non-state-owned firms tend to prioritize market orientation and possess more flexible management mechanisms and decision-making processes [57], allowing them to swiftly adapt to market dynamics and effectively carry out digital transformation initiatives, which in turn drives the development of Nqpf. Moreover, high-tech firms often have strong R&D teams and innovation capabilities [58] and have relatively rich reserves of digital hardware infrastructure. Their higher digitalization level can better facilitate the development of Nqpf; while non-high-tech manufacturing enterprises usually face higher technical barriers and investment costs in digital transformation. Finally, manufacturing enterprises in the growth stage often have organizational flexibility and a willingness to innovate [59], and thus can quickly discover and integrate digital technologies, thereby empowering the development of Nqpf. Manufacturing firms in the decline stage, due to resource scarcity and lack of innovation, find it difficult to support effective transformation, seriously hindering the development of Nqpf.

## 6.2. Theoretical significance

The theoretical contributions of this paper are as follows:

Firstly, this paper examines the impact of the digital transformation of manufacturing enterprises on the development of Nqpf, thereby enriching the existing research on the outcome effects of digital transformation and deepening the industry adaptability of digital transformation in promoting the development of Nqpf. On the one hand, although there is already a wealth of research on the outcome effects of digital transformation, such as its impact on corporate performance [60] and innovation capabilities [61], empirical evidence on how enterprises at the micro – level can cultivate Nqpf through digital transformation and then promote high quality and efficient development remains relatively scarce. On the other hand, most of the existing literature uses listed companies across all industries as samples [62], ignoring the unique attributes of manufacturing enterprises, such as labor intensity and resource dependence. The transformation logic of manufacturing enterprises is fundamentally different from that of other industries.

Secondly, from the perspective of dynamic capabilities, this paper incorporates digital transformation, dynamic capabilities, and Nqpf into a theoretical framework and systematically analyzes how digital transformation promotes the development of Nqpf by enhancing enterprises' dynamic capabilities. Existing studies on the relationship between digital transformation and Nqpf mostly focus on the mediating paths at the enterprise financial or governance levels [63,45], lacking indepth exploration of the transmission mechanism of enterprise capabilities. This perspective provides a new theoretical framework for understanding the relationship between digital transformation and Nqpf and enriches the application scenarios of the dynamic capabilities theory.

Finally, this paper verifies the differential impacts of digital transformation on the Nqpf of enterprises with different characteristics, and provides a feasibility analysis for understanding the role of digital transformation in different enterprises. The effects of digital transformation are often moderated by the characteristics of the enterprises themselves. However, the existing research still lacks indepth exploration of the theory regarding which enterprises can better cultivate Nqpf through digital transformation.

## 6.3. Practical significance

This paper, drawing on the current status of digital transformation and the development of Nqpf in manufacturing listed companies, and in combination with the research conclusions obtained, puts forward the following suggestions:

Firstly, manufacturing firms should align with current trends and promptly capitalize on the critical opportunities presented by digital transformation. By prioritizing the development of smart factories, enterprises can leverage digital technologies such as artificial intelligence and big data analytics to refine and optimize production strategies and inventory management, ultimately driving the intelligent modernization of their manufacturing processes. During this process, the skill transformation of employees is a key support for success. Enterprises must systematically carry out digital skills training and upgrading plans for employees to ensure that employees at all levels can master new tools proficiently and adapt to the working mode in an intelligent production environment, accelerating the pace of digital transformation.

Secondly, manufacturing firms need to focus on the cultivation of dynamic capabilities to foster the development of Nqpf. First, firms can set up an "innovation fund" to encourage employees to propose practical innovative ideas around cost reduction and efficiency improvement or new product development. Second, strengthen the in-depth participation of R&D, production, supply chain and other departments in key projects and decisions, and at the same time, jointly build laboratories with universities and complementary enterprises to break down information barriers and enhance the ability to absorb and integrate new technologies and knowledge. Finally, encourage employees to show a positive and adaptable attitude when facing challenges, strengthen supply chain resilience, and continuously improve the overall adaptability and market competitiveness of the enterprise.

Thirdly, the government ought to establish distinct digital transformation policies to promote the balanced development of Nqpf. When formulating policies, the government ought to appropriately tilt towards enterprises with weak digital transformation levels. First, for state-owned firms, set up a special fund for digital transformation to support state-owned firms in key technology research and application demonstrations in areas such as intelligent manufacturing and industrial internet. For non-high-tech firms, the government should guide them to adopt mature and reliable digital technologies and products, and help improve their production automation and management intelligence levels through policy subsidies and technical training. At the same time, encourage non-high-tech firms to participate in digital transformation pilot projects and explore suitable transformation paths through practice. For firms in the decline stage, reduce their initial investment costs for transformation through interest-subsidized loans and tax credits, and build regional digital transformation public service platforms to provide digital transformation technical consulting services.

### 6.4. Limitations and future research

First, while this study focuses on Shanghai and Shenzhen listed manufacturing firms, over 90% of China's manufacturing sector comprises small and medium–sized enterprises (SMEs) facing unique challenges of "hesitation to transform" and "lack of transformation expertise". The current sample insufficiently captures the transformation characteristics of this critical group. Future research should expand the sample scope to include SMEs, thereby enhancing the study's generalizability. Second, relying solely on annual reports and database data may introduce biases, as these sources lack first-hand production data. Subsequent studies could incorporate workshop-level data collection to improve the reliability of findings. Third, the exclusion of moderating variables may result in context-specific discrepancies. Government policy incentives represent critical drivers of digital transformation in Chinese enterprises. Future models should incorporate policy variables as moderators to clarify the boundary conditions under which digital transformation impacts new quality productive forces development.

### Author contributions

**Conceptualization:** junling wang.

**Data curation:** yongjun zhang.

**Funding acquisition:** junling wang.

**Software:** ruifen zhao.

**Writing – original draft:** junling wang, yongjun zhang.

**Writing – review & editing:** ruifen zhao.

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
