## [Decision Letter · Decision Letter 0]

9 Jul 2025

Dear Dr.. zhao,

Thank you for submitting your manuscript to PLOS ONE. After careful consideration, we feel that it has merit but does not fully meet PLOS ONE’s publication criteria as it currently stands. Therefore, we invite you to submit a revised version of the manuscript that addresses the points raised during the review process.

**Comments from the PLOS Editorial Office** : In order to comply with PLOS's publication criteria requiring that submissions must contribute to the base of academic knowledge, experiments and analyses must be conducted rigorously, with appropriate controls and methods must be described in sufficient detail for others to replicate the analyses (http://journals.plos.org/plosone/s/criteria-for-publication), we request you to address the following additional concerns in your revised manuscript:

1. Please clearly state the source of data for each variable utilized in your manuscript.

2. Please provide a clear definition for each of the variables utilized in your manuscript, such that it is clear how they were derived.

3. Please provide a list of the keywords used for the construction of variables for example, digital transformation.

4. We have noted that the contribution of this manuscript in light of some other related research works such as 10.56028/aemr.11.1.512.2024; 10.1108/imds-09-2024-0907; 10.2991/978-94-6463-570-6_141; 10.3390/su17062652 has not been adequately. Therefore, kindly revise your manuscript to clearly indicate the contribution of this manuscript in light of the related published works identified above.

We thank you for your attention to these requests.

Please submit your revised manuscript by Aug 23 2025 11:59PM. If you will need more time than this to complete your revisions, please reply to this message or contact the journal office at plosone@plos.org . A rebuttal letter that responds to each point raised by the academic editor and reviewer(s). You should upload this letter as a separate file labeled 'Response to Reviewers'.A marked-up copy of your manuscript that highlights changes made to the original version. You should upload this as a separate file labeled 'Revised Manuscript with Track Changes'.An unmarked version of your revised paper without tracked changes. You should upload this as a separate file labeled 'Manuscript'.

We look forward to receiving your revised manuscript.

Kind regards,

Tachia Chin

Academic Editor

PLOS One

[This work was supported by the Hebei Federation of Social Sciences [grant numbers 20230202007]; Hebei Federation of Social Sciences [grant numbers 202307013]; And Shijiazhuang Science and Technology Bureau [grant numbers 245010055A]].

5. Please ensure that you refer to Figure 1 in your text as, if accepted, production will need this reference to link the reader to the figure.

Additional Editor Comments:

After comprehensively considering the comments from the two reviewers, although the paper has certain value in its research topic, and the empirical research quality is acceptable with good performance in technical aspects of quantitative research, there are currently many areas that need improvement. Therefore, the author is requested to comprehensively revise the paper and respond to each reviewer's comments. The revised paper will be sent for re - review, and publication can be considered only after it passes the review.

Reviewers' comments:

Reviewer's Responses to Questions

**Comments to the Author**

1. Is the manuscript technically sound, and do the data support the conclusions?

Reviewer #1: Yes

Reviewer #2: Partly

2. Has the statistical analysis been performed appropriately and rigorously?

Reviewer #1: Yes

Reviewer #2: No

3. Have the authors made all data underlying the findings in their manuscript fully available?

Reviewer #1: Yes

Reviewer #2: No

4. Is the manuscript presented in an intelligible fashion and written in standard English?

Reviewer #1: Yes

Reviewer #2: No

Reviewer #1: Thank you for this opportunity to review your Manuscript. This article investigated the impact of digital transformation in manufacturing industries on new quality productive forces and the mediating role of dynamic capabilities. Although it provides some novel insights, this study still has some shortcomings:

1.Introduction

(1)The definition of the "new quality productive forces " in the first paragraph is rather general. It is suggested to supplement its definition.

(2)It only mentions that dynamic capabilities are a "promising mediators", but does not explain how it specifically act on the relationship between digital transformation and new quality productive forces. It is suggested to supplement this part, explain the reasons for choosing dynamic capabilities, and highlight the importance of dynamic capabilities .

(3)Although the introduction mentions the impact of digital transformation on new quality productive forces, it does not provide a detailed explanation of the current situation, and specific effects of this transformation on the enterprise. This raises questions about the urgency and importance of the research, and makes it uncertain why it is important or what specific gaps it aims to fill in the current knowledge system. Therefore, I suggest specifying the important role of digital transformation and new quality productive forces , and explaining the research gap that the paper aims to fill.

2.Theoretical analysis and research hypotheses

(1)This part has relatively little theoretical support. It is suggested to cite relevant theories to enrich this content and thereby enhance the depth of the paper.

(2)The exposition of the Dynamic capabilities is relatively basic, only mentioning the definition of the Dynamic capabilities, without fully demonstrating the development and application of the theory in subsequent research. Therefore, I suggest citing more relevant research results to enrich this part of the content, supplementing more recent studies that have expanded or validated the theory, in order to reflect the forefront and completeness of the theory, and thus enhance the depth of the theory.

(3)The dynamic capabilities directly adopt the three-dimensional division (innovation, absorption, and adaptation capabilities) of Wang & Ahmed (2007), but it is not explained why this framework was chosen instead of other classic classifications ,such as the three-dimensional division (sense,seize,transform) of Teece (2007).

(4)The specific connotations of "innovation capacity", "absorptive capacity" and "adaptive capacity" have not been clarified. It is suggested to supplement their definitions

3. Research Design

Only "database" was mentioned without specifying the exact name of the database. It is recommended to clarify the source of the data.

4. Conclusion

(1)The literature support for the theoretical exposition in this part is slightly insufficient. It is suggested to supplement relevant literature to enhance the rigor and scientific nature of the argument.

(2)The conclusion section only briefly lists a few relationship results obtained from the research, which seems relatively thin. Therefore, I suggest briefly explaining the findings of this study and explaining the differences and connections between the results of this study and previous research. Combining the analysis results with existing research, further summarizing, improving, and enhancing the conclusion content.

(3) The “Practical significance” section states that “fostering a digital mindset and innovation capabilities”, this is a common practice. Therefore, I suggest proposing specific operational plans.

I suggest the author to improve their manuscript by reviewing and adding recent literature, and finally, I would like to encourage the author to improve their manuscript. The quality of empirical research is good, and quantitative research performs well in terms of technology, which is worth further research. I believe that the above points need improvement I once again congratulate the authors of this study and hope that my comments are constructive and helpful for the development of your manuscript.

Best regards.

Reviewer #2: This paper empirically examines the impact of digital transformation in manufacturing on Nqpf by analyzing panel data from listed manufacturing companies on China's Shanghai and Shenzhen stock exchanges (2011–2022). The study provides theoretical and practical insights into the mechanisms through which digital transformation drives Nqpf. However, the following issues require revision:

1. Refine Hypothesis Development: Beyond existing hypotheses, elaborate on the nuanced dimensions of how digital transformation affects Nqpf, specifically detailing the mediating roles of innovation, absorptive, and adaptive capacities. The current hypothesis development is overly verbose and lacks logical coherence. Each hypothesis must be grounded in explicit theoretical foundations (e.g., innovation theory, resource-based view) with clear reasoning.

2. Strengthen Mediation Analysis & Methodology: While the model references Jiang T's two-step mediation test, the hypotheses inadequately elaborate on how the mediating variables (innovation/absorptive/adaptive capacities) influence Nqpf. Given the limited literature explicitly linking these three specific capacities to Nqpf, it is recommended to adopt a three-step mediation test to mitigate severe endogeneity concerns like spurious regression. Key references include:

[1] Chin, T., Li, Z., Huang, L., & Li, X. (2025). How artificial intelligence promotes new quality productive forces of firms: a dynamic capability view. Technological Forecasting & Social Change, 216.

[2] Tian, H., Zhao, L., Li, Y., Wang, W. (2023). Can enterprise green technology innovation performance achieve “corner overtaking” by using artificial intelligence? —Evidence from Chinese manufacturing enterprises. Technological Forecasting & Social Change, 194, 122732.

3. Enhance Robustness Checks for Endogeneity: The endogeneity treatment section should be strengthened beyond basic tests. Conduct additional robustness checks, such as shortening the sample period or employing alternative regression models, to convincingly demonstrate model stability and reliability.

4. Deepen Heterogeneity Analysis: The current analysis is superficial. Provide meaningful insights by rigorously comparing the effects across diverse firm types (e.g., technology-intensive vs. labor-intensive firms, different lifecycle stages). Identify key differences and commonalities in how digital transformation impacts Nqpf within these groups, offering concrete references for corporate digital policy and practice. For methodology and framing, see:

[1] Chin, T., Li, Z., Huang, L., & Li, X. (2025). How artificial intelligence promotes new quality productive forces of firms: a dynamic capability view. Technological Forecasting & Social Change, 216.

5. Revise Conclusion Section & Enhance Implications: The heading "6. Conclusion" is inadequate; rename it "6. Discussion and Conclusion". Significantly strengthen this section by deriving specific, actionable recommendations for both enterprises (on implementing digital transformation strategies to boost Nqpf) and government policymakers, grounded in the theoretical and empirical findings.

6. Improve Overall Structure, Language, and Formatting:

Optimize the logical flow and transitions between sections for clarity and reader comprehension.Ensure consistent formatting in all tables and figures. For example, maintain three decimal places for Adjusted R² values uniformly (Table 6, Column 2 currently shows four) to enhance professionalism.

Thoroughly polish the English language throughout the manuscript.

**Do you want your identity to be public for this peer review?** For information about this choice, including consent withdrawal, please see our Privacy Policy

Reviewer #1: No

Reviewer #2: No

---

## [Author Response · Author response to Decision Letter 1]

2 Aug 2025

Responses to editor

We sincerely appreciate your valuable suggestions during the review process, which have been of great significance in enhancing the quality of our manuscript. We are particularly grateful for your meticulous attention to the transparency of data sources, the clarity of variable definitions, and the uniqueness of research contributions. These suggestions not only helped us identify the areas for improvement but also deepened our understanding of the rigor and standardization in academic research. We have revised each of your comments and refined the manuscript accordingly. The revised parts are highlighted in yellow in the manuscript. Below are our detailed responses to each of your comments.

1.Comment: Please clearly state the source of data for each variable utilized in your manuscript.

1.Reply: In response to the revision suggestions, in Section 3.1 Sample Selection and Data Sources, we have added detailed explanations of the data sources for the key variables such as digital transformation, new quality productive forces (Nqpf), and dynamic capabilities. We have not only specified the data sources for the specific indicators used to assess the level of Nqpf but also provided the data sources for the three dimensions of dynamic capabilities in enterprises (innovation capability, absorptive capability, and adaptive capability). At the same time, we have also given a detailed account of the data sources for the control variables, ensuring the reproducibility of the research.

2.Comment: Please provide a clear definition for each of the variables utilized in your manuscript, such that it is clear how they were derived.

2.Reply: Thank you for your valuable suggestions. We have clearly defined all the variables. In the first paragraph of the introduction, we defined the concept and characteristics of Nqpf. In section 2.1, " Digital transformation of manufacturing and development of Nqpf", we provided a detailed definition of the concept of digital transformation. In section 2.2, "The mediating role of dynamic capabilities", we also gave detailed definitions of the concepts of dynamic capabilities (innovation capability, absorptive capacity, and adaptability). Additionally, in section 3.2, "Variable Definition and Measurement", we elaborated on the measurement methods for the core variables and the control variables in detail. This explanation ensures that all variables can be derived. We have summarized the measurement methods for each variable in Table 2, "Definition of the main variables", presenting them in a clearer way.

3.Comment: Please provide a list of the keywords used for the construction of variables for example, digital transformation.

3. Reply: Thank you very much for providing such valuable feedback. We admit that our previous work overlooked the mapping of characteristic words for digital transformation. Therefore, we have referred to the research of Wu Fei et al. (2021) and drawn a structured characteristic word map for enterprise digital transformation, as shown in Fig 2. The structured feature-word map of firm digital transformation of the manuscript. This map can clearly illustrate which characteristic words are used to measure the digital transformation of enterprises, thus facilitating the calculation of the degree of digital transformation of enterprises through text analysis.

4.Comment: We have noted that the contribution of this manuscript in light of some other related research works such as 10.56028/aemr.11.1.512.2024; 10.1108/imds-09-2024-0907; 10.2991/978-94-6463-570-6_141; 10.3390/su17062652 has not been adequately. Therefore, kindly revise your manuscript to clearly indicate the contribution of this manuscript in light of the related published works identified above.

4.Reply: Thank you for raising such an important question. We attach great importance to it. First, we found all the four references and compared them one by one. We found that their research fields are roughly the same as ours, so there might be some repetitive parts in the manuscript. We have carried out de-duplication processing to avoid any controversial parts and strictly abide by academic ethics. In addition, for the parts that are of reference value to our research, we have made corresponding citations in the text. We guarantee that there is absolutely no plagiarism and all the materials are original. We also hope that this paper can make certain contributions to the research in related fields.

In addition, in accordance with the requirements of the PLOS ONE journal, we have revised the format of this manuscript and provided an amended funding statement. We have also made available all the original data necessary for replicating the research results and have cited all the figures and tables in the manuscript.

Responses to reviewers

Reviewer # 1:

We sincerely appreciate your review of our manuscript and express our most heartfelt gratitude for your extremely detailed, professional and constructive comments! You not only precisely pointed out the deficiencies in the key parts of our research such as the introduction, theoretical analysis and hypotheses, research design and conclusion, but also provided specific, clear and highly instructive revision suggestions. These comments can help us more clearly recognize the limitations of the current version and the directions that need to be deepened. We fully agree with all your valuable opinions and have carefully and comprehensively revised them, providing detailed responses to each suggestion. The revised parts are highlighted in yellow in the manuscript. The following is a detailed response to each of your comments.

1.Comment: Introduction

(1)The definition of the "new quality productive forces " in the first paragraph is rather general. It is suggested to supplement its definition.

(1)Reply: Thank you very much for your detailed comments. Regarding the issue that the definition of "New quality productive forces (Nqpf)" is rather general, we referred to relevant studies and provided a detailed supplement to its definition in the introduction. Here, the core essence of Nqpf and the characteristics of its three elements are clearly defined.

(2)It only mentions that dynamic capabilities are a "promising mediators", but does not explain how it specifically act on the relationship between digital transformation and new quality productive forces. It is suggested to supplement this part, explain the reasons for choosing dynamic capabilities, and highlight the importance of dynamic capabilities.

(2)Reply: Thank you for your valuable suggestion, which has greatly contributed to the rigor of this paper. When we mentioned that dynamic capabilities are a “promising mediators”, we further elaborated that digital transformation influences dynamic capabilities, including innovation capability, absorptive capability, and adaptive capability. We also explained how dynamic capabilities affect the development of Nqpf. This clarifies how dynamic capabilities specifically function in the relationship between digital transformation and Nqpf, highlighting their significance. This is also a crucial reason why we chose dynamic capabilities as the mediating variable in this paper.

(3) Although the introduction mentions the impact of digital transformation on new quality productive forces, it does not provide a detailed explanation of the current situation, and specific effects of this transformation on the enterprise. This raises questions about the urgency and importance of the research, and makes it uncertain why it is important or what specific gaps it aims to fill in the current knowledge system. Therefore, I suggest specifying the important role of digital transformation and new quality productive forces , and explaining the research gap that the paper aims to fill.

(3)Reply: Thank you for your constructive suggestions. In the introduction, we proposed the significance of developing Nqpf to break through the prominent problems of "low efficiency, high consumption and high emissions" that enterprises still face in the process of transforming from scale expansion to quality and efficiency in the manufacturing industry. However, we found that the overall level of Nqpf in the manufacturing industry is still relatively low, with considerable room for development. Therefore, we put forward that digital transformation is an important factor in promoting the development of new quality productivityNqpf . This explains the urgency and importance of this research. It clarifies that how digital transformation affects Nqpf is not only of great significance for the high-quality development of manufacturing enterprises, but also for the development of advanced manufacturing in various countries. Based on the previous research of scholars, we summarized their limitations. Taking the manufacturing industry as the research object, we aim to fill the gap in empirical literature on the promotion of Nqpf development by digital transformation.

2.Comment: Theoretical analysis and research hypotheses

(1)This part has relatively little theoretical support. It is suggested to cite relevant theories to enrich this content and thereby enhance the depth of the paper.

(1)Reply: Thank you for offering such an important opinion, which has enhanced the rigor of this article. In Section 2.1, "Digital transformation of manufacturing and development of Nqpf", we started from the Resource-Based View to demonstrate that digital transformation is an important foundation for the development of Nqpf. Meanwhile, in Section 2.2, "The mediating role of dynamic capabilities", we cited the dynamic capability theory to prove that dynamic capabilities are an important channel through which digital transformation influences the development of Nqpf. The application of these two theories has further deepened the depth of the paper.

(2)The exposition of the Dynamic capabilities is relatively basic, only mentioning the definition of the Dynamic capabilities, without fully demonstrating the development and application of the theory in subsequent research. Therefore, I suggest citing more relevant research results to enrich this part of the content, supplementing more recent studies that have expanded or validated the theory, in order to reflect the forefront and completeness of the theory, and thus enhance the depth of the theory.

(2)Reply: Thank you for offering such valuable suggestions. In the revised manuscript, we have incorporated more relevant research findings to enrich the development and application of the dynamic capability theory in subsequent studies. This not only demonstrates new research on the role of dynamic capabilities in promoting enterprise performance but also indicates the new expansion of the dynamic capability theory in the field of artificial intelligence. Thus, it reflects the cutting edge and completeness of the dynamic capability theory and further enhances its depth.

(3)The dynamic capabilities directly adopt the three-dimensional division (innovation, absorption, and adaptation capabilities) of Wang & Ahmed (2007), but it is not explained why this framework was chosen instead of other classic classifications ,such as the three-dimensional division (sense,seize,transform) of Teece (2007).

(3)Reply: Thank you for offering such valuable suggestions, which have been of great help in improving this article. In the revised manuscript, we have focused on explaining why we chose the three-dimensional classification of dynamic capabilities proposed by Wang & Ahmed (2007). Considering that digital transformation is a process of adapting to the drastic changes in the external environment such as the impact of digital technology and the upgrading of market demands, and that Nqpf are centered on innovation driven, the three dimensional classification of dynamic capabilities proposed by Wang & Ahmed (2007) is more in line with our research content.

(4)The specific connotations of "innovation capacity", "absorptive capacity" and "adaptive capacity" have not been clarified. It is suggested to supplement their definitions

(4)Reply: Thank you for offering such an important suggestion. In the revised manuscript, we have further clarified the specific connotations of innovation capability, absorption capability and adaptive capability, thereby enriching the research content of this paper.

3. Comment: Research Design

(1)Only "database" was mentioned without specifying the exact name of the database. It is recommended to clarify the source of the data.

(1)Reply: Thank you very much for pointing out the omissions in this article. This was a serious mistake on our part. In the revised manuscript, we have not only specified the exact name of the database but also clearly indicated the source of data for each variable as required by the journal.

4. Comment: Conclusion

(1)The literature support for the theoretical exposition in this part is slightly insufficient. It is suggested to supplement relevant literature to enhance the rigor and scientific nature of the argument.

(1)Reply: Thank you for your constructive suggestions. In each conclusion discussion, we have added relevant literature to enhance the rigor and scientific nature of the arguments. A total of 16 related references have been added.

(2)The conclusion section only briefly lists a few relationship results obtained from the research, which seems relatively thin. Therefore, I suggest briefly explaining the findings of this study and explaining the differences and connections between the results of this study and previous research. Combining the analysis results with existing research, further summarizing, improving, and enhancing the conclusion content.

(2)Reply: Thank you for raising such an important point. This is indeed a weak point of this article. Through this revision, when presenting the empirical result that the digital transformation of the manufacturing industry has significantly enhanced the level of Nqpf, we first explained the research findings, then discussed the differences and connections with previous studies in terms of research objects and the selection of Nqpf indicators, and finally comprehensively discussed and clarified the research significance of this article, highlighting its research contributions. Secondly, when presenting the research result that the innovation ability, absorption ability and adaptability of enterprises play a partial mediating role in the process of promoting the development of Nqpf through the digital transformation of the manufacturing industry, we immediately provided an explanation for this result, then discussed the differences between the mediating mechanisms in previous studies on digital transformation and Nqpfand this article, and once again verified a research conclusion from previous studies. Finally, we comprehensively discussed and clarified the important research significance of taking dynamic capabilities as the mediating mechanism. In addition, when presenting the result that the digital transformation has a greater enhancing effect on the level of Nqpf in non-state-owned and high-tech manufacturing enterprises, we first provided an explanation for this result. Then, by comparing the previous studies and the newly added life cycle heterogeneity test in this article, we discussed the differences and connections with previous research. Finally, we comprehensively discussed and clarified the important research significance of the heterogeneity test.

(3)The “Practical significance” section states that “fostering a digital mindset and innovation capabilities”, this is a common practice. Therefore, I suggest proposing specific operational plans.

(3)Reply: Thank you for your valuable research, which is of great significance to the improvement of the quality of this paper. In Section 6.3, “Practical Significance”, we have provided more specific and feasible suggestions for the development of manufacturing enterprises based on the research conclusions. We have refined the previous broad and vague suggestions, making them more detailed and concrete. For instance, we recommend the adoption of digital technologies such as AI and big data to analyze and optimize production plans and inv

---

## [Decision Letter · Decision Letter 1]

23 Dec 2025

Dear Dr. zhao,

Thank you for submitting your manuscript to PLOS ONE. After careful consideration, we feel that it has merit but does not fully meet PLOS ONE’s publication criteria as it currently stands. Therefore, we invite you to submit a revised version of the manuscript that addresses the points raised during the review process.

We look forward to receiving your revised manuscript.

Kind regards,

Tachia Chin

Academic Editor

PLOS One

Journal Requirements:

**Additional Editor Comments:**

The reviewers appreciate the improvements made; however, they have highlighted several areas that still require attention.

Please revise your manuscript according to the reviewers’ comments, with particular focus on the following points:

1、Introduction – Clarify the connection between general digital technologies (e.g., AI, big data, cloud computing) and the manufacturing field; add specific examples such as Industrial IoT or smart factories.

2、Empirical Analysis – Ensure that all tables are correctly formatted and presented; for instance, Table 5 appears in Chinese in the text.

3、Discussion and Conclusion – Make the theoretical contributions more specific, clearly identify gaps in existing literature, and explain how your study fills them.

Reviewers' comments:

Reviewer's Responses to Questions

**Comments to the Author**

Reviewer #1: All comments have been addressed

Reviewer #3: All comments have been addressed

2. Is the manuscript technically sound, and do the data support the conclusions?

Reviewer #1: Yes

Reviewer #3: Yes

3. Has the statistical analysis been performed appropriately and rigorously?

Reviewer #1: Yes

Reviewer #3: Yes

4. Have the authors made all data underlying the findings in their manuscript fully available?

Reviewer #1: Yes

Reviewer #3: Yes

5. Is the manuscript presented in an intelligible fashion and written in standard English?

Reviewer #1: Yes

Reviewer #3: Yes

Reviewer #1: Thank you for this opportunity to review your Manuscript. This article investigated the impact of digital transformation in manufacturing industries on new quality productive forces and the mediating role of dynamic capabilities. After the last round of revision, the article has been greatly improved, but there are still some shortcomings:

1.Introduction

The terms such as “artificial intelligence, big data and cloud computing” mentioned in the first paragraph are all very macroscopic and general expressions, lacking a close integration with the specific field of “manufacturing”. Adding one or two specific cases or directions of digitalization in the manufacturing field (such as industrial Internet of Things, smart factories) will make the discussion more solid.

2. Empirical analysis

It is not certain whether it is a format issue. The last column of Table 5. Robustness test results in the text is shown in Chinese. It is recommended to check the article.

3.Discussion and conclusion

The description of the theoretical significance section Such as “enriching the research on the antecedent factors of Nqpf and extending the theoretical boundaries of existing digital transformation” seems rather vague and fails to highlight the unique value of the research. Suggestions should be more specific, such as what has been “deepened” and where has been “expanded”. It is necessary to clearly point out the specific gaps or deficiencies in the existing literature in explaining the issue of “how digital transformation affects Nqpf”. For instance, was the micro-mechanism overlooked, or was there a lack of attention paid to the specific field of manufacturing? Then explain how your research fills this gap.

I suggest enriching the article by citing recent and relevant literature .Finally, the above points need improvement and I hope my comments will be constructive and helpful to the refinement of your manuscript.

Best regards.

Reviewer #3: This pape presents an insightful exploration into the impact of digital transformation on new quality productive forces (Nqpf) within the manufacturing industry. The research adopts a novel perspective and employs rigorous scientific methods. Through empirical analysis, the author effectively demonstrates the significant role of digital transformation in enhancing Nqpf, innovatively introducing dynamic capabilities as mediating variables to elucidate their underlying mechanisms. The paper is well-structured, logically coherent, and backed by reliable data sources, offering in-depth analysis. Notably, the theoretical framework, integrating Resource-Based View and Dynamic Capabilities Theory, provides solid theoretical support. Furthermore, the heterogeneity analysis enriches the research content, enhancing the generalizability of the conclusions. This study not only offers theoretical guidance for digital transformation in manufacturing but also provides valuable insights for policymakers, holding substantial academic and practical significance.

**Do you want your identity to be public for this peer review?** For information about this choice, including consent withdrawal, please see our Privacy Policy

Reviewer #1: No

Reviewer #3: No

---

## [Author Response · Author response to Decision Letter 2]

28 Dec 2025

Responses to editor (original comments by editor are in blue color)

We sincerely appreciate the valuable comments you provided during the review process. These comments are of great significance in improving the quality of our manuscript. We are particularly grateful for your meticulous attention to the accuracy of the references and the details of the paper content. These comments not only helped us identify the areas that need improvement, but also deepened our understanding of the rigor and standardization of academic research. We have made revisions based on each of your comments and have accordingly improved the manuscript. The revised parts are highlighted in yellow in the manuscript. Below are detailed responses to each of your comments.

Journal Requirements

1.Comment: If the reviewer comments include a recommendation to cite specific previously published works, please review and evaluate these publications to determine whether they are relevant and should be cited. There is no requirement to cite these works unless the editor has indicated otherwise.

1.Reply: Thank you for your careful reminder. In the first external review opinion, Reviewer 2 mentioned two important references. We carefully read these two papers and indeed found that they are highly relevant to this thesis. When constructing the research model, we referred to the paper by Tian et al. (2023). However, when conducting the test of heterogeneity in the enterprise life cycle, we did not refer to the paper by Chin et al. (2025) provided by Reviewer 2 because its method of dividing the enterprise life cycle was not suitable for this study. Therefore, this thesis referred to the more appropriate research by Li et al. (2011). In the second external review opinion, no expert raised such an opinion.

39.Tian H, Zhao L, Yunfang L, et al. Can enterprise green technology innovation performance achieve “corner overtaking” by using artificial intelligence?—Evidence from Chinese manufacturing enterprises. Technological Forecasting and Social Change. 2023;194:122732. https://doi.org/10.1016/j.techfore.2023.122732

44.Li YH, Li Z, Tang SL. Corporate Life-cycle, Corporate Governance and Corporate Capital Allocation Efficiency. Nankai Business Review. 2011;14(03):110-121. https://doi.org/10.3969/j.issn.1008-3448.2011.03.013

2.Comment: Please review your reference list to ensure that it is complete and correct. If you have cited papers that have been retracted, please include the rationale for doing so in the manuscript text, or remove these references and replace them with relevant current references. Any changes to the reference list should be mentioned in the rebuttal letter that accompanies your revised manuscript. If you need to cite a retracted article, indicate the article’s retracted status in the References list and also include a citation and full reference for the retraction notice.

2.Reply: Thank you very much for your valuable suggestions. According to the journal's requirements, we carefully reviewed the reference list and found no cases of citations of papers that have been withdrawn. We ensured its completeness and accuracy. At the same time, when modifying 6.2 Theoretical significance, we re-quoted 5 new references and highlighted them in yellow at the end of the reference list.

Additional Editor Comments

1.Comment: Introduction – Clarify the connection between general digital technologies (e.g., AI, big data, cloud computing) and the manufacturing field; add specific examples such as Industrial IoT or smart factories.

1. Reply: Thank you for your very constructive suggestion. In the first paragraph of the introduction, we discussed the deep penetration and wide application of advanced technologies such as artificial intelligence, big data, and cloud computing. To enhance the discussion, we included two real examples from manufacturing enterprises. One is that China's Xiaomi Automobile in the automotive manufacturing sector achieved efficient and automated production through an intelligent factory. The other is that the Electronic Works Amberg (EWA) in the electronic manufacturing sector utilized the MindSphere industrial Internet of Things operating system to achieve full-process data collection and intelligent control. Additionally, when exploring the strategic and future industries brought about by digital technologies, we listed the rapid development of embodied intelligent robots and quantum information industries, which are currently popular. Through these examples, we further proved that the digital transformation of manufacturing is rapidly promoting the development of new quality productive forces (Nqpf) of enterprises.

2.Comment: Empirical Analysis – Ensure that all tables are correctly formatted and presented; for instance, Table 5 appears in Chinese in the text.

2.Reply: Thank you for raising such detailed questions. We are extremely sorry that due to our carelessness, such a trivial mistake occurred in the text. We have revised the issue in Table 5 and carefully checked all the formatting of the entire text to ensure that such a trivial error will not occur again.

3.Comment: Discussion and Conclusion – Make the theoretical contributions more specific, clearly identify gaps in existing literature, and explain how your study fills them.

3.Reply: Thank you for your very important suggestion. We also noticed that the previous part about theoretical significance seemed rather vague, failing to highlight the unique value of the research. Firstly, this article explores and enriches the existing research on the outcome effects of digital transformation, and deepens the first theoretical significance of the industry adaptability of digital transformation promoting the development of Nqpf. It explains that the empirical evidence of existing research on how enterprises at the micro level cultivate Nqpf through digital transformation is still relatively scarce. At the same time, existing literature mostly uses samples of all industry listed companies and ignores the unique industry attributes of manufacturing enterprises, thereby highlighting the theoretical value of this study. Secondly, this article explores the significance of the theoretical framework of "digital transformation - dynamic capability - Nqpf", and explains that existing research mostly focuses on the mediating paths at the enterprise financial or governance level, lacking in-depth exploration of the enterprise capability transmission mechanism.Finally, this article explores the differentiated impacts of digital transformation on the Nqpf of enterprises with different characteristics, and explains that existing research still lacks theoretical exploration on which enterprises can better cultivate Nqpf through digital transformation. Therefore, in this revision, we have made the theoretical significance more specific and compared it with existing research, further highlighting the theoretical value of this study.

Responses to reviewers (original comments by reviewers are in blue color)

Reviewer # 1:

We sincerely thank you for your review of our paper, and we express our most heartfelt gratitude for your extremely detailed, professional and constructive comments! You not only accurately pointed out the deficiencies in the key parts of our research (such as the introduction, formatting, and theoretical significance), but also provided specific, clear and highly instructive revision suggestions. These comments can help us more clearly recognize the limitations of the current version and the directions that need to be further deepened. We fully agree with all your valuable opinions and have carefully and comprehensively made the revisions. We have given detailed responses to each suggestion. The revised parts are marked in yellow in the paper. Here are the detailed responses to each of your comments.

1.Comment: Introduction

The terms such as “artificial intelligence, big data and cloud computing” mentioned in the first paragraph are all very macroscopic and general expressions, lacking a close integration with the specific field of “manufacturing”. Adding one or two specific cases or directions of digitalization in the manufacturing field (such as industrial Internet of Things, smart factories) will make the discussion more solid.

1.Reply: Thank you for your very constructive suggestion. In the first paragraph of the introduction, we discussed the deep penetration and wide application of advanced technologies such as artificial intelligence, big data, and cloud computing. To enhance the discussion, we included two real examples from manufacturing enterprises. One is that China's Xiaomi Automobile in the automotive manufacturing sector achieved efficient and automated production through an intelligent factory. The other is that the Electronic Works Amberg (EWA) in the electronic manufacturing sector utilized the MindSphere industrial Internet of Things operating system to achieve full-process data collection and intelligent control. Additionally, when exploring the strategic and future industries brought about by digital technologies, we listed the rapid development of embodied intelligent robots and quantum information industries, which are currently popular. Through these examples, we further proved that the digital transformation of manufacturing is rapidly promoting the development of new quality productive forces (Nqpf) of enterprises.

2.Comment: Empirical analysis

It is not certain whether it is a format issue. The last column of Table 5. Robustness test results in the text is shown in Chinese. It is recommended to check the article.

2.Reply: Thank you for raising such detailed questions. We are extremely sorry that due to our carelessness, such a trivial mistake occurred in the text. We have revised the issue in Table 5 and carefully checked all the formatting of the entire text to ensure that such a trivial error will not occur again.

3. Comment: Discussion and conclusion

The description of the theoretical significance section Such as “enriching the research on the antecedent factors of Nqpf and extending the theoretical boundaries of existing digital transformation” seems rather vague and fails to highlight the unique value of the research. Suggestions should be more specific, such as what has been “deepened” and where has been “expanded”. It is necessary to clearly point out the specific gaps or deficiencies in the existing literature in explaining the issue of “how digital transformation affects Nqpf”. For instance, was the micro-mechanism overlooked, or was there a lack of attention paid to the specific field of manufacturing? Then explain how your research fills this gap.

3.Reply: Thank you for your very important suggestion. We also noticed that the previous part about theoretical significance seemed rather vague, failing to highlight the unique value of the research. Firstly, this article explores and enriches the existing research on the outcome effects of digital transformation, and deepens the first theoretical significance of the industry adaptability of digital transformation promoting the development of Nqpf. It explains that the empirical evidence of existing research on how enterprises at the micro level cultivate Nqpf through digital transformation is still relatively scarce. At the same time, existing literature mostly uses samples of all industry listed companies and ignores the unique industry attributes of manufacturing enterprises, thereby highlighting the theoretical value of this study. Secondly, this article explores the significance of the theoretical framework of "digital transformation - dynamic capability - Nqpf", and explains that existing research mostly focuses on the mediating paths at the enterprise financial or governance level, lacking in-depth exploration of the enterprise capability transmission mechanism.Finally, this article explores the differentiated impacts of digital transformation on the Nqpf of enterprises with different characteristics, and explains that existing research still lacks theoretical exploration on which enterprises can better cultivate Nqpf through digital transformation. Therefore, in this revision, we have made the theoretical significance more specific and compared it with existing research, further highlighting the theoretical value of this study.

Reviewer # 3:

Although you did not offer any comments on this study this time, we are extremely grateful for your previous insightful views on the rigor of the research methods, the depth of the data analysis, and the degree of integration of theory and practice. These have been of great help to us. Once again, we thank you for your recognition of this study. We have further reviewed our paper, including details such as formatting and language, to ensure it meets the requirements for publication. The revised parts are marked in yellow in the paper. Below are detailed responses to each of your comments.

1.Comment: This pape presents an insightful exploration into the impact of digital transformation on new quality productive forces (Nqpf) within the manufacturing industry. The research adopts a novel perspective and employs rigorous scientific methods. Through empirical analysis, the author effectively demonstrates the significant role of digital transformation in enhancing Nqpf, innovatively introducing dynamic capabilities as mediating variables to elucidate their underlying mechanisms. The paper is well-structured, logically coherent, and backed by reliable data sources, offering in-depth analysis. Notably, the theoretical framework, integrating Resource-Based View and Dynamic Capabilities Theory, provides solid theoretical support. Furthermore, the heterogeneity analysis enriches the research content, enhancing the generalizability of the conclusions. This study not only offers theoretical guidance for digital transformation in manufacturing but also provides valuable insights for policymakers, holding substantial academic and practical significance.

1.Reply: We sincerely thank the reviewers for their meticulous review and professional, constructive evaluations of this paper. We also sincerely appreciate your full affirmation of the research topic, analytical perspective, research methods, and academic value of this paper! This paper focuses on the research core of enabling Nqpf development through digital transformation in the manufacturing industry. We attempt to establish an analytical framework integrating the resource-based view and the dynamic capability theory, introduce dynamic capability as an intermediary variable to reveal the internal mechanism of the two, and expand the research boundaries through heterogeneity analysis to enhance the universality of the conclusions. Throughout the research process, we have always strived for the rigor of empirical analysis and the completeness of theoretical logic. Your recognition not only affirms the research ideas and results of this paper, but also points out the direction for our subsequent revision and improvement work. In the future, we will strictly follow the review opinions, further optimize and polish the details of the manuscript, continuously improve the argumentation logic, and solidify the research conclusions, striving to make the research content more profound and rigorous. Once again, we express our most sincere gratitude to the reviewers for your valuable time and professional academic guidance!

---

## [Decision Letter · Decision Letter 2]

29 Jan 2026

Digital transformation, dynamic capabilities and new quality productive forces: Empirical data from listed Chinese manufacturing companies

PONE-D-25-26635R2

Dear Dr. Zhao,

We’re pleased to inform you that your manuscript has been judged scientifically suitable for publication and will be formally accepted for publication once it meets all outstanding technical requirements.

Kind regards,

Tachia Chin

Academic Editor

PLOS One

Additional Editor Comments (optional):

Reviewers' comments:

Reviewer's Responses to Questions

**Comments to the Author**

Reviewer #1: All comments have been addressed

Reviewer #3: All comments have been addressed

2. Is the manuscript technically sound, and do the data support the conclusions?

Reviewer #1: Yes

Reviewer #3: Yes

3. Has the statistical analysis been performed appropriately and rigorously?

Reviewer #1: Yes

Reviewer #3: Yes

4. Have the authors made all data underlying the findings in their manuscript fully available?

Reviewer #1: No

Reviewer #3: Yes

5. Is the manuscript presented in an intelligible fashion and written in standard English?

Reviewer #1: Yes

Reviewer #3: Yes

Reviewer #1: After two rounds of revision, the article has been greatly improved. I would like to thank the authors for their responses to my comments and for the thorough revision of the manuscript. I believe the manuscript can be published now.

Reviewer #3: Accept. This is a good work, they have addressed my concerns. This paper have meet the publication quality.

**Do you want your identity to be public for this peer review?** For information about this choice, including consent withdrawal, please see our Privacy Policy

Reviewer #1: No

Reviewer #3: No

---

## [Editor Report · Acceptance letter]

PONE-D-25-26635R2

PLOS One

Dear Dr. zhao,

I'm pleased to inform you that your manuscript has been deemed suitable for publication in PLOS One. Congratulations! Your manuscript is now being handed over to our production team.

Kind regards,

on behalf of

Dr. Tachia Chin

Academic Editor

PLOS One